# DD-CISENet: Dual-Domain Cross-Iteration Squeeze and Excitation Network for Accelerated MRI Reconstruction

**Xiongchao Chen**[1,2]                                                XIONGCHAO.CHEN@YALE.EDU

**Zhigang Peng**[1]                                      ZHIGANG.PENG@SIEMENS-HEALTHINEERS.COM

**Gerardo Hermosillo Valadez**[1]                  GERARDO.HERMOSILLOVALADEZ@SIEMENS-
HEALTHINEERS.COM

[1] *Siemens Healthineers, Malvern, PA 19355, USA*

[2] *Department of Biomedical Engineering, Yale University, New Haven, CT 06511, USA*

**Editors:** Under Review for MIDL 2023

## Abstract

Magnetic resonance imaging (MRI) is widely employed for diagnostic tests in neurology. However, the utility of MRI is largely limited by its long acquisition time. Acquiring fewer k-space data in a sparse manner is a potential solution to reducing the acquisition time, but it can lead to severe aliasing reconstruction artifacts. In this paper, we present a novel **D**ual-**D**omain **C**ross-**I**teration **S**queeze and **E**xcitation **Net**work (DD-CISENet) for accelerated sparse MRI reconstruction. The information of k-spaces and MRI images can be iteratively fused and maintained using the Cross-Iteration Residual connection (CIR) structures. This study included 720 multi-coil brain MRI cases adopted from the open-source fastMRI Dataset (Zbontar et al., 2018). Results showed that the average reconstruction error by DD-CISENet was $2.28 \pm 0.57\%$, which outperformed existing deep learning methods including image-domain prediction ($6.03 \pm 1.31\%$, $p < 0.001$), k-space synthesis ($6.12 \pm 1.66\%$, $p < 0.001$), and dual-domain feature fusion approaches ($4.05 \pm 0.88\%$, $p < 0.001$).

**Keywords:** Deep learning, MRI reconstruction, dual-domain, multi-coil parallel imaging

## 1. Introduction

Magnetic resonance imaging (MRI) is an essential clinical diagnosis tool of neurology. However, the long scanning time of MRI might induce many problems including patient discomfort, high exam cost, and motion artifacts. One potential approach for accelerated MRI scanning is downsampling k-space measurements. However, the reconstructed images using the downsampled k-space data will display severe aliasing artifacts.

Deep learning has shown great potentials in the accelerated sparse reconstruction of MRI. Existing deep learning approaches can be generally classified into three categories. The first category applied the sparsely reconstructed MRI images as input of neural networks to predict the synthetic fully reconstructed images. The second category utilized the sparse k-space as input of networks to generate the synthetic full-view k-space. The third category combines the features of k-space and images in a dual-domain manner to restore the full-view k-space (Eo et al., 2018). However, the cross-iteration features were typically ignored in previous dual-domain methods. In this study, we present a novel **D**ual-**D**omain **C**ross-**I**teration **S**queeze-**E**xcitation **Net**work (DD-CISENet) for the accelerated sparse reconstruction of brain MRI. The incorporated Cross-Iteration Residual (CIR) connections enable data fusion across iterations to enhance the reconstruction accuracy.

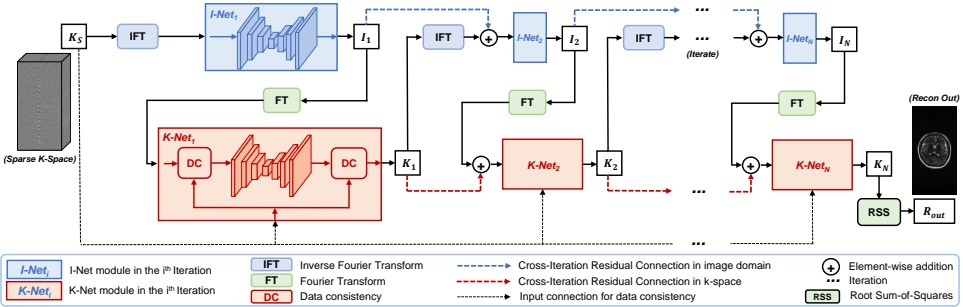

Figure 1: The architecture of DD-CISENet. *I-Net* or *K-Net* modules are end-to-end connected for dual-domain feature fusion. Cross-Iteration Residual (CIR) connections enable the retention of image or k-space features across iterations.

## 2. Methods

The diagram of DD-CISENet is presented in Fig. 1. The sparse k-space data $K_S$ is first input to *I-Net*$_1$ module after reconstruction using Inverse Fourier Transform (IFT), generating the output $I_1 = \mathcal{H}_{I_1}(\mathcal{F}^{-1}(K_S))$, where $\mathcal{H}_{I_1}$ refers to a dual Squeeze-Excitation Network (SENet) (Chen et al., 2021) in *I-Net*$_1$. $\mathcal{F}^{-1}$ refers to the IFT operator.

Then, $I_1$ was input to the *K-Net*$_1$ module after forward projection using Fourier Transform (FT), generating the output $K_1$. Then, the IFT of $K_1$ is input to *I-Net*$_2$ of the $2^{nd}$ iteration. Meanwhile, $I_1$ is also added to *I-Net*$_2$ using CIR connections to produce the output $I_2 = \mathcal{H}_{I_2}(\mathcal{F}^{-1}(K_1) + I_1)$. Thus, the image-domain features of the $1^{st}$ iteration is retained and transmitted to the next iteration to better incorporate the image features.

Similarly, $K_1$ is added to *K-Net*$_2$ to retain the k-space features. The output k-space of the i$^{th}$ (i≥2) iteration can be formulated as:

$$K_i = \mathcal{D}(\mathcal{H}_{K_i}(\mathcal{D}(\mathcal{F}(I_i) + K_{i-1}, K_S)), K_S), \tag{1}$$

where $\mathcal{D}$ is a data consistency module. $\mathcal{H}_{K_i}$ is SENet in *K-Net*$_i$. $\mathcal{F}$ is the FT operator. Then, the predicted $K_N$ is reconstructed into the final MRI image $R_{out}$ as the output.

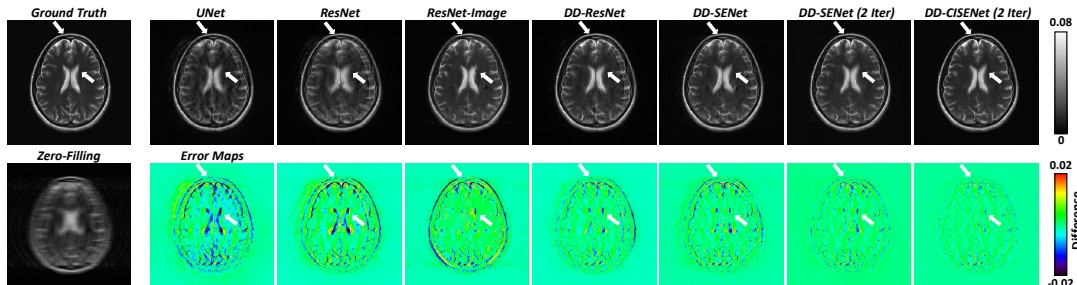

Figure 2: Visualizations of the reconstructed MRI images using predicted k-space. Difference maps are placed at the bottom side for comparison.

The overall end-to-end loss function of DD-CISENet is $\mathcal{L} = \mathcal{L}_1 + \cdots + \mathcal{L}_i + \cdots + \mathcal{L}_N$, where $\mathcal{L}_i = \lambda_{I_i} l_{I_i} + \lambda_{K_i} l_{K_i}$ represents the combined loss of the $i^{th}$ iteration. $\lambda$ is the loss

weight that was empirically set as 0.5 in our study. $l_{I_i}$ or $l_{K_i}$ is the $L2$ loss between the ground-truth fully sampled and predicted images or k-spaces.

## 3. Results and Conclusions

Fig. 2 shows the qualitative comparison of the reconstructed MRI images by multiple approaches. The proposed *DD-CISENet* outputs more accurate MRI images than existing image-domain prediction techniques (*ResNet-Image*), k-space synthesis methods (*UNet*, *ResNet*), and dual-domain feature fusion approaches (*DD-ResNet*, *DD-SENet*). Table 1 lists the quantitative comparison of the generated k-space and reconstructed MRI images by multiple approaches using normalized mean square error (NMSE) and structural similarity (SSIM). It can be observed that *DD-CISENet* presents quantitatively more accurate k-space data and reconstructed MRI images than existing methods. Paired t-tests further validated the statistical significance of the quantification results ($p < 0.001$). Thus, the proposed DD-CISENet demonstrated state-of-the-art performance in MRI sparse reconstruction, superior to existing image-domain, k-space, and dual-domain methods.

Table 1: Quantification of the generated k-space data and reconstructed images.

| Methods | Generated K-space Data | | Reconstructed MRI Images | | |
|---|---|---|---|---|---|
| | NMSE% | SSIM | NMSE% | SSIM | P-value[†] |
| Zero-Filling | $19.27 \pm 6.05$ | $0.075 \pm 0.001$ | $13.65 \pm 4.21$ | $0.984 \pm 0.005$ | − |
| UNet | $10.32 \pm 2.25$ | $0.132 \pm 0.027$ | $6.97 \pm 1.41$ | $0.991 \pm 0.003$ | $< 0.001$ |
| ResNet | $8.06 \pm 2.14$ | $0.151 \pm 0.026$ | $6.12 \pm 1.66$ | $0.991 \pm 0.002$ | $< 0.001$ |
| ResNet-Image | − | − | $6.03 \pm 1.31$ | $0.990 \pm 0.001$ | 0.202 |
| DD-ResNet | $7.50 \pm 2.25$ | $0.183 \pm 0.036$ | $4.05 \pm 0.88$ | $0.992 \pm 0.001$ | $< 0.001$ |
| DD-SENet | $7.08 \pm 2.61$ | $0.186 \pm 0.021$ | $3.52 \pm 0.88$ | $0.995 \pm 0.001$ | $< 0.001$ |
| DD-SENet (2 Iter) | $6.85 \pm 2.26$ | $0.207 \pm 0.034$ | $2.49 \pm 0.66$ | $0.997 \pm 0.001$ | $< 0.001$ |
| DD-CISENet (2 Iter) | $\mathbf{6.54 \pm 2.56}$ | $\mathbf{0.221 \pm 0.034}$ | $\mathbf{2.28 \pm 0.57}$ | $\mathbf{0.998 \pm 0.001}$ | $< 0.001$ |

[†]P-value of paired t-test on NMSE of Images between the current and previous group.

## Acknowledgments

This work was funded by Siemens Medical Solutions USA, Inc.

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
