# OpenReview forum: "DD-CISENet: Dual-Domain Cross-Iteration Squeeze and Excitation Network for Accelerated MRI Reconstruction"
_MIDL.io/2023/Short_Paper_Track — MIDL 2023 Short paper track Poster_

### Official Review · Reviewer_PW9t · 2023-04-25
**Review of: DD-CISENet: Dual-Domain Cross-Iteration Squeeze and Excitation Network for Accelerated MRI Reconstruction**

**Rating:** 8
**Confidence:** 4

**Review:**

The authors present a joint image-space and k-space (recurrent) processing network for accelerated MR image reconstruction, presenting results on accelerated head MR.

Overall this abstract is quite good, and I have few notes. However, I would encourage the authors to compare and contrast their method to
Singh et al. 2022, Joint Frequency and Image Space Learning for MRI Reconstruction and Analysis
which also features an image/k-space processing.

In the initial transfer between image and k-space after I-net 1, should there be a residual computation between observed data back-filled (means or zeros) and FFT[ output of I-net 1 ]? While the back-filling surely is sub-optimal, compared to the next steps which have some form of residual operation before input to the next K-net this input step will clearly have scale differences. Should the K-Net operation (and the corresponding I-Net operation) be expected to handle these scale shifts?

Or, alternatively, should the input to the k-net be multi-channel here?

As it stands the results are good, so these questions are purely curiosity, but I think testing differences between residuals/addition and learned combinations in the initial layer may be interesting. Others have noted similarities to iterative denoising methods and recurrent networks (see, e.g., Regularization by Denoising), and these often have reliance on a weighted sum (convex combination) of the iterates. Adding them directly might actually be the critical case without convergence, so it's not unreasonable to instead try sum weights in converging regime...however, since the 2-iteration case is the only one used in practice, this is only a "theory gain".

---

### Official Review · Reviewer_z7ZM · 2023-04-25
**MRI reconstruction by joint analysis of image and frequency data**

**Rating:** 4
**Confidence:** 4

**Review:**

This paper addresses the problem of undersampled MRI reconstruction. The authors propose a neural architecture that combines frequency and image space data to perform reconstruction. The paper falls short on discussion of relevant work and comparison to the state of the art in this very active area of research that also includes methods that operate jointly in image and frequency space. The authors compare results to some very simple baselines rather than state of the art methods, thus leaving the reader to wonder if the proposed method offers any advance on quality of reconstruction.

Some examples of relevant work:

TaejoonEo,YohanJun,TaeseongKim,JinseongJang,Ho-JoonLee,andDosikHwang. KIKI-net:Cross-domain convolutional neural networks for reconstructing undersampled magnetic resonance images. Magnetic resonance in  medicine,80(5):2188–2201,2018.

Roberto Souza and Richard Frayne. A hybrid frequency-domain/image-domain deep network for magnetic resonance image reconstruction. In2019 32nd SIBGRAPI Conference on Graphics, Patterns and Images (SIBGRAPI),pages257–264.IEEE,2019.

Bo Zhou and S Kevin Zhou. DuDoRNet: Learning a dual-domain recurrent network for  fast MRI reconstruction with deepT1prior.InProceedingsoftheIEEE/CVFConferenceon ComputerVisionandPatternRecognition,pages4273–4282,2020.

Nalini M Singh, Juan Eugenio Iglesias, Elfar Adalsteinsson, Adrian V Dalca, Polina Golland. Joint frequency and image space learning for MRI reconstruction and analysis. The Journal of Machine Learning for Biomedical Imaging, 2022-018, p 1-28, 2022.